# Kin Selection in the RNA World

**DOI:** 10.3390/life7040053

**Published:** 2017-12-05

**Authors:** Samuel R. Levin, Stuart A. West

**Affiliations:** Department of Zoology, University of Oxford, South Parks Road, Oxford OX1 3PS, UK; stuart.west@zoo.ox.ac.uk

**Keywords:** RNA cooperation, kin selection, RNA world, Hamilton’s rule, limited diffusion, origin of the genome, scale of competition, modelling the origin of life

## Abstract

Various steps in the RNA world required cooperation. Why did life’s first inhabitants, from polymerases to synthetases, cooperate? We develop kin selection models of the RNA world to answer these questions. We develop a very simple model of RNA cooperation and then elaborate it to model three relevant issues in RNA biology: (1) whether cooperative RNAs receive the benefits of cooperation; (2) the scale of competition in RNA populations; and (3) explicit replicator diffusion and survival. We show: (1) that RNAs are likely to express partial cooperation; (2) that RNAs will need mechanisms for overcoming local competition; and (3) in a specific example of RNA cooperation, persistence after replication and offspring diffusion allow for cooperation to overcome competition. More generally, we show how kin selection can unify previously disparate answers to the question of RNA world cooperation.

## 1. Introduction

Life very likely began as simple replicating RNA molecules [1,2,3]. These first replicators were capable of little more than making copies of themselves. However, the last universal common ancestor already contained a complex genome, wrapped inside a cell, capable of varied metabolic and replicative tasks. Replicators in the RNA world then had many obstacles to overcome. Molecules had to successfully copy themselves and each other. Different kinds of ribozymes, such as polymerases and synthetases, had to evolve and stably persist. Independent replicators had to come together to form the first genomes. Each of these steps involved various biochemical hurdles, and most of these biochemical puzzles remain largely unsolved.

However, in addition to posing biochemical problems, many of the key steps in the evolution of the RNA world are also problems of cooperation [4,5,6,7]. For example, in various plausible RNA world scenarios, molecules act as enzymes to increase the replication rate of other molecules (Figure 1). This can be considered a cooperative trait, because by acting as enzymes, these molecules reduce their own replication rate to help copy others. A selfish mutant that receives the benefits of the enzymatic activity of others, but does not act as an enzyme itself, would have higher fitness. Consequently, all else being equal, we expect selfish molecules to outcompete cooperative ones. The problem is simple: Why would a replicating molecule help copy others instead of selfishly copying itself as fast as it could? Similar problems arise in synthestases cooperatively producing nucleotides and independent replicators coming together to form the first genomes. Each of these issues is a different problem of cooperation in the RNA world, and a number of explanations have been put forward to resolve these problems, including limited diffusion, primitive cells, and spiral waves [5,6,8,9,10,11,12,13,14,15,16,17].

The links between these different suggested solutions for the problem of cooperation in the RNA world are not clear. Are they different explanations, or can we, instead, identify an overarching framework that links them all? It is useful here to make a comparison of the literature on the evolution of cooperation in higher organisms, ranging from animals to bacteria. Over the last 50+ years, work in this area of ‘social evolution’ has produced relatively unified theoretical and empirical literature that can explain cooperation across the tree of life [18,19,20]. One theory that could be especially relevant to the RNA world is kin selection [6,9,17]. Hamilton showed that cooperation can be explained if it is directed towards relatives [21]. Natural selection favours genes that are better able to get copies of themselves into the next generation. Hamilton’s kin selection theory highlights that a gene can get a copy of itself into the next generation by either increasing the replication of the individual it is in, or by increasing the replication of other individuals that carry copies of that gene.

Kin selection requires that cooperation be directed towards relatives. While this can involve mechanisms to discriminate kin from non-kin, it can also work via limited dispersal keeping relatives together. For example, when bacteria grow clonally, a cell will be tend to be surrounded by genetically identical cells, facilitating cooperation. This could potentially be important in the RNA world if limited diffusion keeps copies of the same molecule together (Figure 2). Indeed, several of the models developed to explain cooperation in the RNA world appear similar to previously developed kin selection models. If cooperation in the RNA world can be explained by kin selection, then this would simplify our picture of the world, unifying existing RNA models and showing how the same process can drive biochemical and organism level evolution.

Our aim is to test the utility of using existing kin selection methodologies to explain cooperation in the RNA world. We use a game theory approach to determine under what conditions cooperation would be evolutionarily stable, and hence be favoured by natural selection. This approach is deliberately simple, abstracting away biological details, to focus on key parameters that are likely to be important across different systems. We start with the simplest possible model, and then elaborate by adding in details that could have been important in the RNA world. Our more specific questions are: (1) Why would cooperation have been favoured in the RNA world? (2) How will different RNA biochemistries influence the evolution of cooperation? (3) Can we gain anything from applying the kin selection approach to the RNA world?

## 2. Results

### 2.1. A Simple Model of RNA Cooperation

We start by developing a simple model of cooperation in RNA molecules using standard kin selection techniques [22]. We deliberately avoid tying the model to a specific RNA system, keeping it general to capture a broad range of possible systems. A similar approach has recently been taken with viruses [23]. We imagine that RNA molecules have some potentially cooperative trait that benefits the other RNA molecules with whom they are interacting (social partners). For example, an RNA replicator might act as a cooperative enzyme, increasing the replication rate of the local replicators. However, at the same time, this cooperation comes at a cost to the individual performing the cooperation, reducing their replication rate. For example, a cost could arise because acting as an enzyme reduces the amount of time a molecule is available in template form for other molecules [16,24]. Consequently, there is a trade-off here, with cooperation benefiting the group, but being costly to the individual.

In this case, the replication rate, or fitness, of an RNA molecule will be a function of its own level of cooperation (the individual cost) and the local level of cooperation amongst the RNA molecules it is interacting with (the group benefit). We assume that the focal RNA molecule has phenotype *x* and that the average phenotype of the social partners it is interacting with is *y*. The phenotype could represent the likelihood (bounded between zero and 1) or amount (unbounded) of cooperation. For example, *x* could be the probability, between 0 and 1, of becoming a cooperative enzyme. A simple way to model the costs and benefits of cooperation is to assume that our focal RNA molecule has a baseline replication rate of $W_{b}$, which will be reduced by some function depending upon its own level of cooperation (*C*), and increased by some function of the level of cooperation in the local group of RNA molecules (*B*). The replication rate, *w*, can then be expressed as:
(1)$$w\left(x,y\right)=W_{b}-C\left(x\right)+B\left(y\right).$$


We can think of *C* as the cost of the cooperative trait and *B* as the benefit of the trait. Equation (Equation 1) is analogous to other models that have looked at cooperation both generally and in specific systems such as microbes [22,25,26,27,28,29]. We now ask what strategy would be favoured by natural selection. More formally, we seek the evolutionarily stable strategy (ESS) [30]. An ESS is a strategy ($x^{*}$) for which, if all individuals in the population express it, no rare mutant variant will have a higher replication rate (Figure 3). The ESS approach focuses on phenotypes, and assumes that, within a range set by the modeller, all phenotypes are possible. For example, the likelihood that an RNA molecule cooperates could be between 0 and 1. Another way of thinking about this is that the ESS approach looks at which direction evolution would proceed by looking for the unbeatable strategy. 

Candidate ESSs occur where fitness is maximized. Taylor and Frank (1996) showed that, assuming weak selection, this happens when the derivative of fitness with respect to genotype is zero [31]:
(2)$$\frac{dw}{dg}=\frac{\partial w}{\partial x}+R\frac{\partial w}{\partial y}=0.$$


*R* is genetic relatedness, which is a measure of genetic similarity [21,32,33]. What does relatedness mean in the context of RNA replicators? In general, relatedness captures the likelihood that two individuals share genes, and therefore it measures an individual’s vested interest (in an evolutionary sense) in others. Genetic similarity between partners can come about a number of ways. Thus, *R* is a very general parameter which captures all processes that generate genotypic associations between individuals.

In the case of RNAs, this association could come about, for example, through limited diffusion of offspring copies. Limited diffusion leads to identical copies finding themselves near each other, meaning *R* is high. Consequently, a focal molecule’s genetic sequence can become better represented in the population either by copying itself or its neighbours, which are likely to be identical. In the RNA world, relatedness usually measured the proportion of interactants that had an identical sequence, but relatedness was able to arise from any heritable correlation in RNA traits. As an example, in the simplest case of a social group containing τ individuals with identical phenotypes and π with different phenotypes, $R=\frac{\tau }{\tau \;+\;\pi }$ (for τ individuals). This simple measure of relatedness allows us to capture many different possible configurations of genotypes and phenotypes in a single parameter, vastly simplifying our analysis. In this model *R* is equal to $\frac{dy}{dx}$, and relatedness is a measure of phenotypic correlation—the more closely related you are to individuals, the more similar your phenotypes will be.

Solving Equation (Equation 2) for $x=y=x^{*}$ (a monomorphic population) gives the value of the ESS, $x^{*}$. The condition for cooperation to evolve is $x^{*}>0$ (because x=0 would be no cooperation). For Equation (Equation 1), assuming baseline fitness is 1, cooperation evolves when
(3)$$RB^{\prime }-C^{\prime }>0,$$
where $B^{\prime }=\frac{\partial w}{\partial y}$ is the benefit of cooperation, $C^{\prime }=\frac{\partial w}{\partial x}$ is the cost of cooperating, and *R* is relatedness.

Equation (Equation 3) tells us that RNA cooperation will be favoured if the marginal cost of cooperation is smaller than the marginal benefit of cooperation, weighted by relatedness between social partners. Thus, Equation (Equation 3) captures previous results that cooperation can be favoured by limited diffusion, spiral waves, or primitive cells. Limited diffusion and spiral waves lead to identical copies finding themselves near each other, which generates high *R* [7,17]. Primitive cells generate high *R* by keeping identical individuals together from one generation to the next [9].

Equation (Equation 3) also illustrates how we can think about cooperation in RNA molecules as being favoured by kin selection. If there is a high likelihood that molecules will interact with identical molecules (high R), then cooperation will more readily evolve. Specifically, Equation (Equation 3) is a classic result known as Hamilton’s (1964) rule, which has been used to explain cooperation across the tree of life [21,28,32]. In particular, a role for limited dispersal in generating a positive relatedness, and hence favouring cooperation by kin selection, has been demonstrated in a range of organisms, including bacteria, slime moulds, insects, birds and mammals [34,35,36,37,38,39,40,41,42].

### 2.2. Different Types of RNA Cooperation

In the above section, we were deliberately vague about the functions *B* and *C*, to keep them general. We did this so that the model would capture a wide range of potentially cooperative traits in the RNA world. However, different types of RNA molecules will engage in different types of cooperation, and it can be useful to consider these more explicitly. One issue is that sometimes cooperators can also be recipients of the benefits of cooperation and other times they are not. For example, a cooperative polymerase might replicate a nucleotide synthetase, leading to more nucleotides in the environment, which will benefit the focal polymerase as well as all those around it. Alternatively, by acting as a replicase to increase the replication rate of others, a replicator might no longer be able to be copied itself, such that it receives none of the benefits of cooperation. Do these different forms of cooperation lead to different evolutionarily stable strategies? Does cooperation evolve to the same degree regardless of the form it takes?

We can answer these questions by extending the previous model to explicitly model the nature of the cooperative trait. We again imagine an RNA replicator can express some cooperative traits, like acting as an enzyme to increase the replication rate of others, at a cost to itself. We use the term ‘cooperative’ in line with other work in the field, but since in this case replicators incur a lifetime fitness cost to provide a benefit to others, this is ‘altruism’ in the strict sense [43]. An individual’s strategy relates to the proportion of its offspring copies that express the cooperative phenotype. For example, alternate folding patterns would allow different copies to express different phenotypes. A more cooperative strategy would be one where a higher proportion of offspring acts as cooperative enzymes.

Baseline fitness is assumed to be 1. We take 0≤x≤1 to be the proportion of a focal individual’s offspring copies that cooperates. For example, if x=1, all the copies of an offspring act as cooperative enzymes (complete cooperation). If x=0, none of its offspring cooperate (complete selfishness). Values between 0 and 1 are considered partial cooperation. *b* and 0≤c≤1 are the benefits and costs, respectively, of cooperating (where here, for simplicity, we have substituted concrete effects on replication rate for the functions in Equation (Equation 1)). For example, if an individual becomes an enzyme that helps replicate others, it loses *c* from its baseline replication rate, but increases the replication rate of others around it by a factor *b*.

To distinguish between different forms of cooperation, we incorporate a new parameter, 0≤β≤1. β measures the degree to which, given a molecule is cooperative, it can still receive the benefits of cooperation (by). For example, if folding to act as an enzyme prevented a cooperator from being replicated by other enzymes, this would be represented by β≃0. Or if the cooperator mined some public good, like nucleotides, which benefits the group, this would lead to β≃1, as the cooperator can receive the benefits of other molecules mining nucleotides (Figure 4).

An individual’s fitness will be the sum of the replication rates of the fraction *x* of its offspring which act as cooperators and the fraction (1−x) that is selfish:
(4)w(x,y)=x(1−c+βby)+(1−x)(1+by).


Offspring copies that do not act as cooperators have, on average, a 1+by relative replication rate. Offspring that act as cooperators have a 1−c+βby replication rate, where β measures the degree to which cooperators receive benefits. The Taylor–Frank method shows that the direction of selection is given by
(5)$$\frac{dw}{dg}=1-c+\beta bx-1-bx+R\left(\beta bx+\left(1-x\right)b\right).$$


First, we consider the extreme case where being a cooperator has no effect on an RNA’s ability to receive benefits, or β=1. In game theoretic terms this is equivalent to an additive game. This fits the scenario, for example, in which a molecule can ‘mine’ nucleotides from the environment, which benefits all individuals in the group, including the cooperator. In this case, Equation (Equation 5) reverts to Equation (Equation 3), and cooperation will evolve when
(6)Rb−c>0,
which again represents a simplified form of Hamilton’s rule. Otherwise, if β>0, candidate ESSs are given by
(7)$$x^{*}=\frac{Rb-c}{b(1+R)-\beta b(1+R)}.$$


Here and for all subsequent analyses we assume that $x^{*}$ is bounded between 0 and 1 (checking that the boundaries are stable when this is not true). Equation (Equation 7) is a general result for a negatively synergistic game played by RNA molecules in which being a cooperator reduces a replicator’s ability to be a recipient of cooperation. We find that:
Regardless of whether cooperative RNAs receive benefits (the value of β), the condition for cooperation to evolve ($x^{*}>0$) is Rb−c>0. This tells us that the degree to which cooperators receive benefits has no effect on *whether* cooperation will evolve, although it impacts on the degree of cooperation. Regardless, higher benefits and relatedness and lower costs are favourable for RNA cooperation, confirming the more general model in the previous section.In the extreme case in which cooperative RNAs receive no benefits (β=0), the ESS value of cooperation is capped at 0.5, and only partial cooperation can evolve, because complete cooperation would mean that there were no individuals available to receive benefits. This applies, for example, in some RNA trans-replicases, in which becoming a replicase enzyme prevents individuals from being replicated by other replicases [16,44]. This result is analogous to the result obtained in Frank’s (1997) model of paired sibling suicide in animals [45]. This suggests that the evolution of the cooperative enzyme that cannot receive benefits (as in [16,24,44]) is analogous to the evolution of sterility in higher organisms.


More generally, β determines the degree to which complete cooperation can evolve in RNAs (Figure 5), and provides a parameter that applies across forms of cooperation in the RNA world. Because RNA molecules are so simple, and phenotypes will often be expressed through folding patterns, we expect low β to be common in the RNA world. Therefore, we expect complete cooperation to be rare amongst RNAs.

### 2.3. Cooperation and Competition in RNA World

We have shown above that positive genetic relatedness can help favour cooperation (Equations (Equation 3), (Equation 6), and (Equation 7)). However, we have implicitly assumed that relatives can be together for cooperative interactions without also competing with each other. This might not always be the case. RNA molecules may compete for resources, like nucleotides or binding sites on a surface. Whether or not this selects against cooperation can depend on biochemical details.

For example, limited diffusion of molecules (e.g., in [13,16,46]) leads to relatives being near each other (high relatedness). However, it also leads to the individuals with which a replicator competes also being relatives. In that case, competition is relatively *local*, as the extra individuals produced by competition impact the local group. Alternatively, replicators might act locally to increase each other’s replicative rates, but disperse in a vesicle to compete *globally*. For example, abiotic protocells formed from amphilic molecules could divide by shearing and combine with each other [47]. In this case, the extra copies produced by cooperation displace individuals at a global level. Exactly how do relatedness and competition interact and impact cooperation?

We can answer this question by extending the previous model (Equations (Equation 4)), taking the case of β=0 for simplicity) to incorporate competition, with a new parameter, 0≤a≤1. *a* measures the scale of competition, with a proportion *a* of competition occurring in the local social group, and the remaining 1−a occurring globally [22,34,48]. For example, if a=0, all competition occurs at the population level, with each individual competing equally with every other individual. If a=1, all competition occurs within the local social group—extra offspring produced only displace local individuals. Because we are now distinguishing between local and global competition, we must distinguish between the average phenotype of the social group (*y*) and the average phenotype of the population ($\bar{y}$):
(8)$$\begin{array}{cc} w(x,y,\bar{y}) & =\frac{f}{aF+(1-a)F}\\ & =\frac{x(1-c)+(1-x)(1+by)}{a\left(y\left(1-c\right)+\left(1-y\right)\left(1+by\right)\right)+\left(1-a\right)\left(\bar{y}\left(1-c\right)+\left(1-\bar{y}\right)\left(1+b\bar{y}\right)\right)}. \end{array}$$


The term in the numerator is a focal replicator’s fitness (*f*), which is a relative denominator that measures the average fitness the focal individual competes against. The denominator is composed of the local (*F*) and global (F) average fitnesses, where the relative importance of each is determined by *a*.

The fitness function in Equations (Equation 8) is analogous to Takeuchi et al.’s model of the origin of genome-like molecules, which looked at the evolution of template-like, selfish molecules, and protein-like cooperative molecules from a single starting point (though we only capture the evolutionary, not ecological aspects of their model) [24]. In their model, replicators that act as cooperative catalysts (protein-like molecules) cannot also act as templates (DNA-like molecules) β=0, and mutations vary the probability with which a replicator acts as one or the other.

The Taylor–Frank method gives the ESS to be
(9)$$x^{*}=\frac{Rb-c-a(Rb-Rc)}{b(1+R)-a2Rb}.$$


Equation (Equation 9) tells us that:
When competition is completely global (a=0), the ESS reverts to the that identified in the previous model (Equation (Equation 7), β=0), and the condition for cooperation to evolve is simply Rb−c>0. This confirms our previous results, and is a qualitatively similar result to that found by Takeuchi et al., as in their model competition is relatively global and replicators cooperate about half the time (Equation (Equation 7), [24]).As competition becomes more local (*a* increases), the condition for cooperation to evolve becomes more stringent. When competition (*a*) is high, the competitive effects of the extra offspring copies produced by, e.g., cooperative enzyme activity, are experienced locally. This means that in addition to benefiting from cooperation, relatives suffer increased competition from cooperation, making it harder for cooperation to evolve (Figure 6).When competition is completely local (a=1), $x^{*}=-\frac{c}{b}$. Since the costs and benefits are both positive, cooperation cannot evolve.


We expect many RNA biologies to lead to local competition. This is because most RNA population structures that have been described involve simple diffusion, which does not afford many opportunities for exporting the benefits of cooperation globally. If this is the case, we should be on the lookout for either: (1) different forms of RNA dispersal, other than simple limited diffusion, which could decrease local competition; or (2) other features of RNA biology that might achieve the same effect. We provide an example of the latter in the next section.

The above analysis also links RNA world with the wider literature on the influence of local competition. Previous work on bacteria and insects has demonstrated how local competition can reduce selection for cooperation [39,49,50]. Furthermore, local competition can also select for harmful or spiteful traits that reduce the fitness of competitors [48,51,52,53]. We might expect to find such traits in the RNA world. 

### 2.4. Explicit Population Structure: Closing the Model

In our analysis above we implicitly assumed that *R* is independent from the other model parameters. In reality this is unlikely to be the case. For example, limited diffusion leads to both local competition *and* higher relatedness, and so a and r should be positively correlated [34,35,54,55].

We can allow for this by modelling an explicit RNA life history, the parameters of which can then be used to calculate an estimate of relatedness in system. Incorporating specific life history parameters and using them to calculate relatedness in the model is called ‘closing the model’. This is an alternative to our previous models, which assumed independence between *R* and model parameters (‘open’ models) [54,56]. We develop an explicit population structure model known as an infinite island model. In an island model, an infinite population is subdivided into patches, or groups of individuals, of size *N*. Island models are standard in evolutionary biology, but distinct from the lattice approach often used in RNA models, in that we do not explicitly track distance. While this island model approach is taken for analytical tractability, it has been shown to give qualitatively similar results to explicit lattice structures [57].

We extend the model of cooperation in Equation (Equation 5) by explicitly incorporating two life history parameters: offspring diffusion and parent survival. We expect both parameters to impact relatedness and competition. Offspring remaining nearby, and parents remaining between generations both increase relatedness, but to different degrees, also impacting the scale of competition. An explicit model can determine the full impact of each of these factors.

We assume parent RNA molecules produce offspring copies, and then a proportion of parents, *k*, survives into the next generation, and the remaining 1−k fraction of parents dies. After reproduction, a proportion, *d*, of offspring copies diffuses elsewhere, while the other 1−d fractions remain locally. An individual’s fitness can now be decomposed into the probability it survives and has fitness 1, or dies, and therefore its fitness is a function of whether its offspring diffuse or remain. If offspring diffuse they compete globally, and if they remain they compete locally, against the patch average fitness. We can write fitness as a function of the focal individual’s phenotype, *x*, the average phenotype of the other individuals on the patch, *y*, and as the whole-group average (including the focal individual), *Z* ($\frac{y(N\;-\;1)\;+\;x}{N}$) (see Appendix A for derivation):
(10)$$\begin{array}{cc} w(x,y,Z)= & (1-k)(d)(x(1-c+by)+(1-x)(1+by))+\\ & \left(1-d\right)\frac{x(1-c+by)+(1-x)(1+by)}{1+(1-d)(bZ(1-Z)-cZ)}+k. \end{array}$$


Note that the RNA system captured by the fitness function in Equation (Equation 10) is analogous to Shay et al.’s model of trans-replicases, in which two complementary strands of a trans-replicase can act as (selfish) templates or (cooperative) replicases, which increase the replication rate of templates [16,17]. We do not explicitly track the different phenotypes of the complements, instead looking at the overall probability a replicator is a cooperator (e.g., replicase) or selfish (e.g., template), and looking at the evolution of this probability.

In the Appendix A, we use the Taylor–Frank method to identify the ESS in terms of the model parameters and *R*. We then calculate an estimate of the equilibrium value of *R* in terms of the model parameters by writing a recursion for how population parameters change *R* from one generation to the next. Once the equilibrium value of *R* is determined, we substitute back in for *R* to get an ESS of
(11)$$x^{*}=\frac{2bk(1-d)-c(2k(1-d)+N(2+k-d-k(1-d)))}{b(1+k)N-b(1-d)(k(N-4)-N)}.$$


Equation (Equation 11) is a result we previously attained in a model of RNA trans-replicases [17]. It tells us that:
Cooperation is favoured by higher parent survival (larger *k*), limited diffusion (lower *d*), and smaller local group size (smaller *N*).Both parent survival (k>0) and limited diffusion (d<1) are required for cooperation to evolve. While discrete generations (k=0) hold approximately for many higher organisms, we expect overlapping generations (k>0) to hold for most RNA systems, because we expect RNAs to survive well after their copies’ copies have replicated. This offers a potential explanation for why RNA molecules might have cooperated despite having a dispersal strategy that otherwise leads to high local competition.


The approach we have used in this section is a closed modelling approach, where the relatedness emerged from the population parameters of the model (diffusion, survival, etc.), rather than being an open parameter. The benefit of this approach is that it reveals the exact relationship between model parameters and cooperation. The downside is that it required being explicit about the life history and population structure of the RNA molecules. In situations in which we know these details this approach will be useful. Otherwise it may be useful to keep the model open, and subsume unknown population processes in *R*.

## 3. Discussion

We have used the analytical methods of kin selection to model cooperation in the RNA world. We started with a deliberately simple model, abstracting away biochemical details to identify a general process by which cooperation is favoured in the RNA world. We showed that cooperation in RNAs, like many other organisms, is favoured by positive genetic relatedness (our *R*) ( Equation (Equation 3)). Genetic relatedness can arise a number of ways, such as active kin discrimination, or just limited dispersal (population viscosity). Previous explanations for RNA cooperation include limited diffusion, spiral waves, and primitive cells, all of which serve to generate high genetic relatedness (Table 1). We have shown that these previously disparate explanations can be unified under the single explanatory framework of kin selection.

We elaborated on our most simple model by incorporating specific biological details that might be especially relevant in RNA world. We examined whether cooperative RNAs can receive the benefits of cooperation ( Equation (Equation 4)). We expect that it will be common for RNA cooperators to be unable to receive the full benefits of cooperation, which suggests that complete cooperation will be rare in the RNA world (Figure 5) [4,16,17,24]. We then modelled the scale of competition in the RNA world (Equation (Equation 8)). The simple RNA population structures that generate high relatedness, which favours cooperation, are also likely to lead to high local competition, which we have shown disfavours RNA cooperation (Figure 6) (Table 1). This suggests that other life history features are likely to be important for overcoming local competition [17]. We explored this possibility in our final model ((Equation (Equation 10)), by examining whether cooperation can evolve under different conditions of offspring diffusion and parent survival. We showed that for a simple RNA system, both limited diffusion and parent survival (overlapping generations) are necessary for the evolution of cooperation ((Equation (Equation 11)).

We have made a number of assumptions that are standard in kin selection analyses, and it is worth questioning their validity for the RNA world. One issue is that we have assumed that all phenotypes in the strategy set (e.g., 0≤x≤1) are possible. With very simple RNAs this may not be the case, as the phenotype space may not be continuous. In this case, we would need to limit the strategy set, for example to a number of discrete strategies [30,58]. Another issue is that we have assumed that near-equilibrium viable mutant variants have mutations of small effect (weak selection) [21,33]. However, again, the simple nature of RNA molecules may mean that mutations tend to be of large effect [4]. In this case an explicit population genetic model could be more appropriate. Whether these two assumptions are borne out, and how our predictions would change if they were violated, remains to be investigated both theoretically and empirically.

### 3.1. Why Bother?

We have shown that we can think of various types of cooperation in the RNA world as being driven by kin selection, and that we can use social evolution tools to model this process. However, an obvious question arises: If one can also model these processes using other tools such as simulations, why bother with the kin selection methodologies that we have used here? We suggest three main benefits.

#### 3.1.1. Generality

First, the simple analytical nature of these models offers biological insight and generally applicable conclusions. For example, we generated models that focused on the effects of dispersal, cooperation type, or the scale of competition. This approach lends itself to systems in which specific biochemical details are obscured, as is the case in the RNA world, because we do not yet know what the first replicators looked like. Streamlined models isolate key parameter relationships and identify important general biological features, which allows us to extend our conclusions beyond specific systems. Further, we have found that our models make similar predictions to more explicit simulations (e.g., Equation (Equation 9) and [24] or Equation (Equation 11) and [16]), which means that what we gain in generality is not necessarily lost in predictive power. Finally, when very different approaches lead to the same predictions, we gain confidence in those predictions.

#### 3.1.2. Testability

Second, kin selection has been useful for creating a link between theory and experiments in higher organisms, and it should offer the same for the RNA world [20,59]. Kin selection models often identify simple relationships between parameters and traits, that can be tested with both experiments and across-species comparative studies (e.g., [36,37,38,39,40,41,42,59,60,61]).

For example, Figure 5 could be tested through comparative work. Different replicators synthetised in the lab will have different conformational properties: some will be more or else readable by a polymerase when folded to act as a cooperative enzyme, or will be more else able to fold and unfold and therefore express different phenotypes. Equation (Equation 7) makes a simple, testible prediction about the amount of cooperation we expect to see in these different replicators. Figure 6 could be tested experimentally in the lab (experimental evolution), by manipulating the scale of competition, as has been done with bacteria [39]. For example, solutions of RNA replicators could be well mixed (global competition, low relatedness), growing on surfaces (local competition, high relatedness), or growing on surfaces but migrating distantly on the surface of beads (high relatedness, global competition), and Equation (Equation 9) predicts the level of cooperation we expect to evolve under each of these conditions. More generally, this approach identifies relatedness as an important parameter to manipulate experimentally [39,41,49,50,62]. These are a handful of examples, but the link to kin selection provides a wealth of empirical examples to draw from (see Table 1), and simple parameters to test comparatively or experimentally.

Further, kin selection provides a simple heuristic to frame our thinking, generate verbal predictions, and identify fruitful evolutionary problems in the RNA world, which are all advantages for the empiricist. Take, for example, the evolution of a replicase enzyme which copies template strands (e.g., in [16]). We do not have to think of this as cooperation being driven by kin selection. However, doing so identifies life history features that are likely to be important (e.g., diffusion and survival), points us to problems that may arise that have been well studied in other taxa (e.g., the link between competition and relatedness), and makes it easy to generate predictions that can be tested in the lab (a system of replicators with limited diffusion and parent survival should evolve cooperation, whereas a system with limited diffusion alone should not). As origin of life experimental capabilities become more advanced, kin selection could become an increasingly useful tool for guiding empirical work and making testable predictions, as has been the case with animals and microbes [19,20,26,59,63,64].

#### 3.1.3. Conceptual Links

Finally, the kin selection approach allows us make conceptual links to other taxa, unifying our evolutionary explanations across the tree of life. The tools we have used here are the same as those used to study cooperation in bacteria, birds, and mammals [59,61], and we can use them to identify common factors favouring cooperation across taxa. One advantage of this approach is that that we can use insight from the vast existing kin selection literature to guide our thinking about the RNA world. For example, our model of competition (Equation (Equation 8)) identified spite as a potentially important trait to consider in the RNA world. This approach also has the advantage of simplifying our understanding of life, providing a unifying framework rather than generating new explanations for each case. This last advantage is a key goal of science. A number of simulation studies have already demonstrated the utility of more complex, explicit approaches to solving problems in the RNA world (reviewed by [7]). We are not saying that the RNA world must be conceptualised using kin selection, just that it can be useful to do so.

## Figures and Tables

**Figure 1 life-07-00053-f001:**
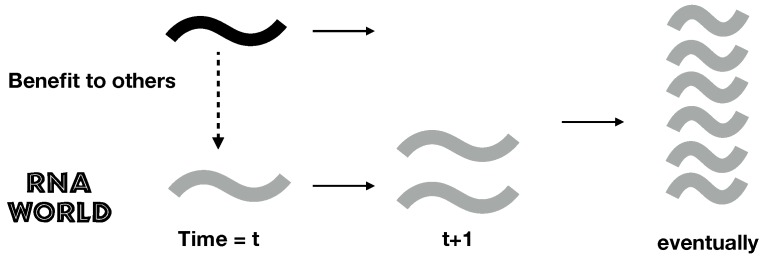
The problem of cooperation in the RNA world. A cooperator (black squiggle) provides a benefit to other individuals (grey squiggle), increasing their relative replicative success at a cost to their own relative success. Over time, all else being equal, individuals that do not incur this cost but receive the benefits have higher replicative success, or fitness, and become better represented in the population. How, then, does cooperation evolve?

**Figure 2 life-07-00053-f002:**
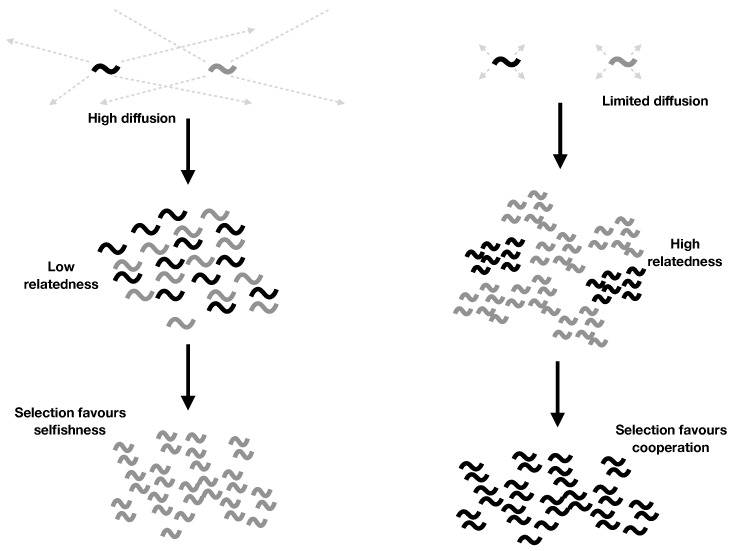
Limited diffusion in the RNA world. Cooperative RNAs are depicted as black squiggles, and selfish ones as grey squiggles. High diffusion leads to a well mixed population (low relatedness), which favours the evolution of selfishness. Limited diffusion leads to high relatedness. Cooperators are more likely to encounter other cooperators, and selfish individuals are unlikely to encounter cooperative ones to exploit. Selection favours cooperation.

**Figure 3 life-07-00053-f003:**
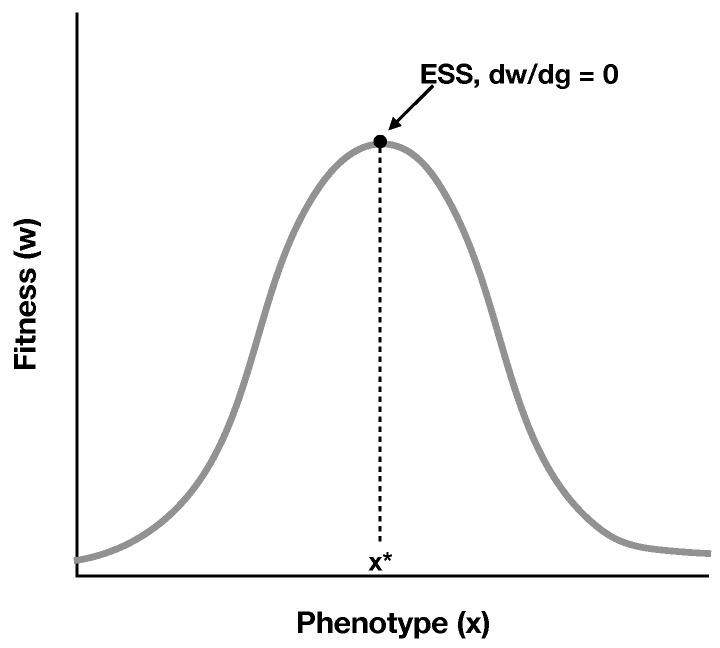
Visual representation of the evolutionarily stable strategy (ESS) approach. Taylor and Frank (1996) developed an approach for identifying ESSs. An equation for fitness (y-axis) as a function of phenotype (x-axis) is either derived or assumed. Natural selection will move populations towards fitness peaks. At a fitness optimum, small phenotypic variations in either direction will have lower fitness, and therefore the population will remain the same. The ESS is the phenotype ($x^{*}$) where this occurs, and for a continuously differential fitness function, this happens at dw/dx=0. We expect organisms to express ESSs as a result of natural selection over time.

**Figure 4 life-07-00053-f004:**
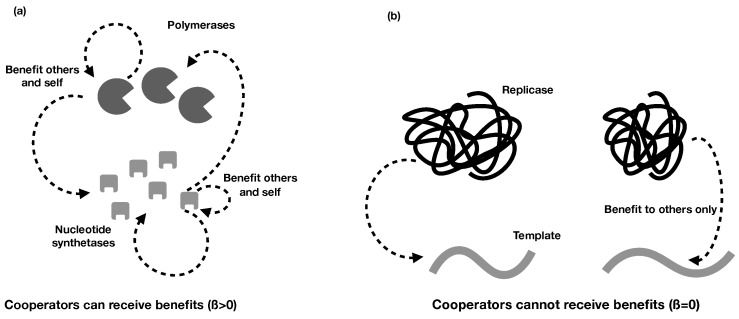
An illustration of the different types of cooperation in the RNA world. (**a**) Nucleotide synthetases (light grey cuboids) make nucleotides which benefit themselves, other nucleotides, and polymerases (dark grey spheroids). β is relatively high, because being a cooperator does not limit a synthetase’s ability to benefit from cooperation. Similarly, polymerases can copy nucleotides and other polymerases, and they receive benefits both by making more synthetases (which leads to more nucleotides) and by being copied by other polymerases. (**b**) A cooperative replicase can copy a template, but cannot be copied by other replicases. Thus, β=0, because being a cooperator prevents one from receiving any benefits from cooperation.

**Figure 5 life-07-00053-f005:**
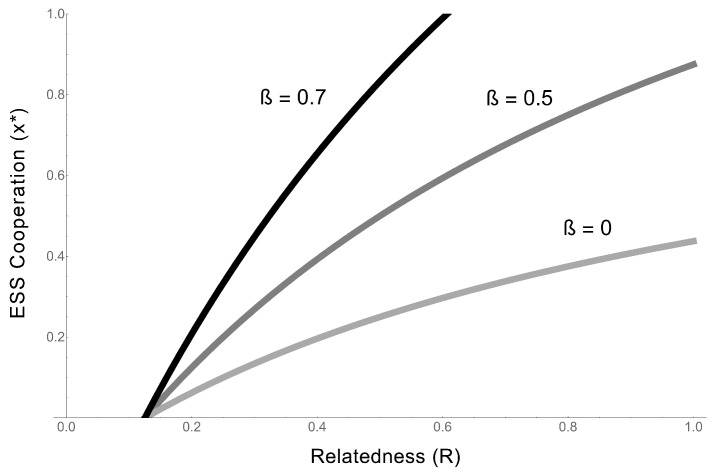
Do cooperators receive the benefits of cooperation? The y-axis shows the ESS level of cooperation ($x^{*}$), plotted against relatedness (*R*). The three lines represent three different values of β, which measures the degree to which cooperators receive benefits. When β is high, cooperators are equally as likely to receive the benefits of cooperative acts as non-cooperators. When β is low, acting as a cooperative RNA limits or prevents a molecule from receiving the benefits of cooperation. When β is low, only partial ($x^{*}<<1$) cooperation can evolve. For all values of β, increasing relatedness (*R*) increases the ESS value of cooperation. For all lines b=0.8, c=0.1.

**Figure 6 life-07-00053-f006:**
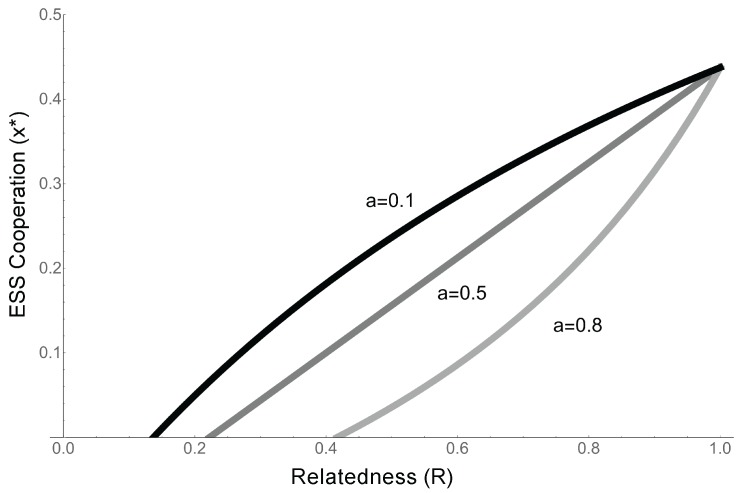
The scale of competition in the RNA world. The y-axis shows the ESS, $x^{*}$, derived from the model in the text for different parameter values. The x-axis is relatedness (*R*). The three lines represent three different values of *a*, the scale of competition. When *a* is low, competition is relatively global. When it is high, competition is relatively local. Increasing *a* reduces the ESS value of cooperation. For all values of *a*, increasing relatedness favours cooperation. For all lines b=0.8, c=0.1.

**Table 1 life-07-00053-t001:** Applying the kin selection approach to example RNA world systems. Rows show different example RNA world model systems, as well as references that have modelled such systems. The columns show four questions which can be asked of a model system: How is relatedness generated? Do cooperators receive benefits? What is the scale of competition? How can the predictions made by such models be tested? Subcolumns (e.g., limited diffusion, comparative) give the answers to those questions. The X symbol shows which of the subcolumns apply to the model system in a given row.

Example Model System	Relatedness Is Generated by			Do Benefits Return to Cooperators? (β)		Scale of Competition (*a*)		Testable Predictions?	
	Limited Diffusion	Proto-Cells	Other Spatial Clustering	Yes (High β)	No (Low β)	Global (Low *a*)	Local (High *a*)	Comparative	Experimental
Replicases in protocells [5,9,15]		X		X		X			X
Replicases on surfaces [12,13]	X			X			X		X
Trans-replicases [16]	X				X		X	X	X
Nucleatase and polymerase [46]	X			X			X	X	X
Origin of genome-like molecules [24]			X	X			X		X
Hyper-cycles [8,11]			X	X			X		X

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
