# Peer review of "Kin Selection in the RNA World"

_life, 2017, doi:10.3390/life7040053_

Round 1

Reviewer 1 Report

It is surely important to understand behaviors of RNAs, such as polymerases and synthetases, in the RNA world, using kin selection, which the authors developed. So, I would like to know whether the authors can answer the following question. How could RNAs or polymerases and synthetases accumulated in the world, acquire genetic information for protein synthesis? However, it may be difficult to answer the question. If it is the case, the authors should give at least speculations on the question. Otherwise, I worry about that the key selection for solving the riddle on the origin of life may become meaningless. 

Author Response

It is surely important to understand behaviors of RNAs, such as polymerases and synthetases, in the RNA world, using kin selection, which the authors developed. So, I would like to know whether the authors can answer the following question. How could RNAs or polymerases and synthetases accumulated in the world, acquire genetic information for protein synthesis? However, it may be difficult to answer the question. If it is the case, the authors should give at least speculations on the question. Otherwise, I worry about that the key selection for solving the riddle on the origin of life may become meaningless. 

***We agree with reviewer 2 that the question of how information begins to accumulate in the world — in other words, how the selective process began — is a fundamental question for origin of life researchers. However, this is not the question our paper addresses. Instead, we are interested in how cooperation evolved once individuals capable of passing on information and undergoing natural selection appeared, because cooperation was likely a key hurdle for many later steps, which we outline in the introduction. Almost by definition the question reviewer 2 raises cannot be answered by evolutionary models, though a number of biochemists are tackling the problem. 

Reviewer 2 Report

This manuscript, by Levin and West, describes a model of RNA cooperation, and it attempts to use it to explain whether cooperative RNAs (defined generally as an RNA that provides some benefit to another RNA at the expense of diminished capacity for replication) benefit from such cooperation, the scale of this cooperation (only in very general, relative terms - i.e., local vs. global), and the effects of diffusion and parent survival during offspring's generation on this model. The authors posit based on their modeling that RNA cooperation is beneficial and operates at the "Kin Selection" level, i.e., that populations of related RNAs are selected as a whole. While an interesting hypothesis - and a plausible one - it's quite unclear how their work can be related to a real-world system. The word "polymerase" occurs only four times in the manuscript. While this is a somewhat trivial criticism in isolation, it is representative of the paper, which generally doesn't consider how the theory might be applied to an experimental system, or even a model system in which idealized catalysts and polymers perform specific behaviors. The authors argue that the simplicity and low-assumption nature of the derivation of the model is an asset, but ultimately, it must be applicable to a real-world system to be of value. In the introduction, the authors state: "in a specific example of RNA cooperation, parent survival and offspring diffusion allow for cooperation to overcome competition." Here, my expectations were raised, then lowered: the "specific example" doesn't refer to specific RNAs, or even specific hypothetical RNAs with idealized behaviors, but rather just the specific variables of parent survival and offspring diffusion. As written, the paper almost doesn't need to be about RNA, but could rather be describing one of the other population-level systems discussed. The work should be tied into the RNA world with specific examples from the literature. The addition of a specific RNA-world related model would significantly strengthen the manuscript and extend its utility to beyond the theoretical community to experimentalists. This doesn't necessarily have to be reduced to a chemically tractable system. A couple nice examples of theoretical work that employ systems with defined behavior without using specific molecules are those described by Higgs (PLOS Comp Bio 2016) and Hud (PLOS One 2012) - this is representative of the sort of thing I'm thinking of. At least ascribing a behavior to an RNA or suite of RNAs and treating them quantitatively using the model, even if it's not an experimental molecular system, would add a great deal of value. Specific criticism: Page 6, Line 194-195: "Alternatively, replicators might act locally to increase each other’s replicative rates, 195 but disperse in a vesicle to compete globally." This doesn't make sense - in a protocell model, the vesicle is the local environment, with the lipid bilayer separating it from the wider environment. There is subcellular localization in extant cells, but this does not appear to be being invoked here. This should be clarified. The whole idea of local vs. global is underdeveloped with regard to the specific molecular system the authors envision. While this is a theoretical paper - in the end, it is still about molecules of RNA, and the authors should describe exactly to what system they anticipate applying the model.

Author Response

This manuscript, by Levin and West, describes a model of RNA cooperation, and it attempts to use it to explain whether cooperative RNAs (defined generally as an RNA that provides some benefit to another RNA at the expense of diminished capacity for replication) benefit from such cooperation, the scale of this cooperation (only in very general, relative terms - i.e., local vs. global), and the effects of diffusion and parent survival during offspring's generation on this model. The authors posit based on their modeling that RNA cooperation is beneficial and operates at the "Kin Selection" level, i.e., that populations of related RNAs are selected as a whole. While an interesting hypothesis - and a plausible one - it's quite unclear how their work can be related to a real-world system. The word "polymerase" occurs only four times in the manuscript. While this is a somewhat trivial criticism in isolation, it is representative of the paper, which generally doesn't consider how the theory might be applied to an experimental system, or even a model system in which idealized catalysts and polymers perform specific behaviors. The authors argue that the simplicity and low-assumption nature of the derivation of the model is an asset, but ultimately, it must be applicable to a real-world system to be of value. In the introduction, the authors state: "in a specific example of RNA cooperation, parent survival and offspring diffusion allow for cooperation to overcome competition." Here, my expectations were raised, then lowered: the "specific example" doesn't refer to specific RNAs, or even specific hypothetical RNAs with idealized behaviors, but rather just the specific variables of parent survival and offspring diffusion. As written, the paper almost doesn't need to be about RNA, but could rather be describing one of the other population-level systems discussed. The work should be tied into the RNA world with specific examples from the literature. The addition of a specific RNA-world related model would significantly strengthen the manuscript and extend its utility to beyond the theoretical community to experimentalists. This doesn't necessarily have to be reduced to a chemically tractable system. A couple nice examples of theoretical work that employ systems with defined behavior without using specific molecules are those described by Higgs (PLOS Comp Bio 2016) and Hud (PLOS One 2012) - this is representative of the sort of thing I'm thinking of. At least ascribing a behavior to an RNA or suite of RNAs and treating them quantitatively using the model, even if it's not an experimental molecular system, would add a great deal of value. 

****We thank reviewer 3 for this feedback — we believe the changes it led to have greatly improved the paper. Specifically, in l.189-196 and l. 226-232, we have added explicit links to existing idealised models of RNA world systems. In l. 286-291 we now describe how our model captures the biological features of one of Higgs’ specific models and discuss in l. 359-361 how our approach makes similar predictions.

We have also added two relevant subsections to the discussion. First, a subsection on generality, which discusses explicitly why having such general models that are not tied to a specific molecule are useful. Second, a subsection on testability, which discusses how our models point to future work and could be tested comparatively and in the lab. 

Finally, we have added Table 1 (l. 394) which makes links between our models and specific systems developed in previous work.

Specific criticism: 

Page 6, Line 194-195: "Alternatively, replicators might act locally to increase each other’s replicative rates, 195 but disperse in a vesicle to compete globally." This doesn't make sense - in a protocell model, the vesicle is the local environment, with the lipid bilayer separating it from the wider environment. There is subcellular localization in extant cells, but this does not appear to be being invoked here. This should be clarified. The whole idea of local vs. global is underdeveloped with regard to the specific molecular system the authors envision. While this is a theoretical paper - in the end, it is still about molecules of RNA, and the authors should describe exactly to what system they anticipate applying the model.

***Thank you, we realise that this was not clear. We have clarified the kind of biology that could lead to global competition in l. 208-213, including a reference to work showing that this kind of biology is plausible in the RNA world.  Importantly, any specific RNA system will only have one value for ‘a’, but we now clarify in Table 1 (l. 394) which example models systems have what relative values for ‘a’. We also have added to the discussion a section on generality, which discusses why having a model which includes something like ‘a’ as a parameter is useful for work on the RNA world, and a section on testability, which shows how comparative work could test the prediction of our model. 

Reviewer 3 Report

This paper provides a basic overview of how kin-selection thinking might benefit discussions surrounding the RNA World and the origins of life. The authors suggest that, by framing our thinking in terms of kin selection, we are better able to bring discoveries from other corners of evolutionary biology to bear on the problems presented by the RNA World. The take-home message is that kin-selection theory is one possible means by which advances can be achieved.

The authors’ modelling starts from the reasonable premise that early replicators were faced with a dilemma. They could either boost the replication rate of others through enzymatic activity, or allow their own replication to be boosted by acting as a template. Increased amounts of time spent boosting others is viewed as more “altruistic,” while increased amounts of time being boosted is viewed as more “selfish.” I understand that “altruism” would need to be present in sufficient quantities in order fuel the hypercycles one might be interested in studying.

The modelling continues by employing kin-selection methods to study simple models for the evolution altruism---i.e. the fuel for the origin of life. The authors’ kin-selection analysis produces three clear results in very short order (that results are easy to derive is a strength of the approach). Specifically,

(1)    The extent to which a replicator can share in the effects of its own altruism will be an important determinant of the evolution of altruism.

(2)    The extent to which replicators within a lineage compete with one another for resources will be another important determinant, with increased self-shading hindering altruism.

(3)    The details of how replicators move around in the environment will further complicate 1 and 2.

Results 1-3 will be familiar to those already well-versed in kin-selection theory, but the fact that they are formulated in terms of the RNA World bridges an important gap, in my view. Given the Aims of the journal, I believe readers will find the contribution interesting.

Still, like any reviewer in the life sciences, I think the submission can be improved. To that end, I’ve offered suggestions for improvement, below.

Main Suggestions:

A.      The models used in the paper are likely to be viewed by the readership as being non-standard. The authors acknowledge this in at least one place (line 241). To really sell the idea that kin-selection theory can make a valuable contribution, I think it is important to apply it to models that will be more familiar to the Life community. I will concede that my suggestion may not be practicable at this stage, and that the absence of an application to lattice-based populations is not a fatal weakness. Bringing the methodology to bear on a model that this community really cares about is certainly something the authors should tackle soon, though.

B.      I agree with the authors that there is something to be gained by viewing the RNA world through the kin selection lens. However, I don’t think they have made a solid case for this. Instead, they simply make vague assertions like that aroung line 333. Specifics, there, are important given the goal of the paper. Can the authors provide an example of how kin-selection thinking might point to new horizons in RNA World work? Another missed opportunity (but not the only one) occurs at line 178. There, the authors mention a connection to some work by Steve Frank, but they don’t elaborate. Bridging the gap is going to require more than simply providing a citation!

C.      A bit more detail on what the kin-selection methodology is about might be useful given the context. After poring over the paper, a reader could be forgiven if he/she claimed that kin-selection is just about swapping out derivatives for relatedness coefficients. But it’s way more than that! For example, the authors may wish to describe kin-selection methodology as a tool for reducing the state space of the evolving system.  By this I mean that the kin-selection methodology uses relatedness coefficients to effectively group different micro-states that, themselves, capture different configurations of genotypes. In other words, one needn’t track all the details about which genes go where; instead one can just use R to average. The basic idea of reducing the number of dimensions of a dynamical system would likely be familiar to a crowd used to seeing lattice-based models.

Minor Suggestions:

Line 40: Hamilton reference is the wrong style.

Line 66: “avoiding”

Line 76: I did not get a good sense of what x was at this point, nor did I realize that it was bounded between 0 and 1 until Line 304.

Line 96: “Relatedness captures the likelihood that two individuals share genes.” I am having difficulty understanding this for RNAs. Are related RNAs simply identical sequences? Please clarify. Also this might be a convenient place to address C, above.

Line 104. I don’t agree that R is formally equivalent to dy/dx. Using R in place of dy/dx is a little heuristic introduced by Taylor and Frank (1996). Formally, R is a covariance divided by a variance (Michod & Hamilton, 1980).

Line 106: “ … for cooperation to evolve …” Is the idea that, without cooperation, there is no life? Can you clarify why x*>0 is particularly important?

Line 150. I’m having a difficult time with the term “cooperative enzyme,” because I keep thinking we’re talking about cooperative binding like with Hb (are we?). If “cooperative enzyme” really refers to a replicator with greater x, then is it better to use “altruistic.” I know that “altruism” has gone out of style, but it might be useful to resurrect it here to avoid other kinds of confusion.

Line 150: delete the ref to equation 4, and make it clear that beta is a number between 0 and 1.

Around eqn 7: Make it clear that this is a *candidate* ESS (you’ve only satisfied the first-order, necessary condition).

Figure 2: this plot illustrates eqn 7 but only in the case where R is independent of x and beta. As you say later in the paper, R might depend on these things (and possibly more). I think that discussion needs to come earlier in the paper; otherwise Figure 2 might be a bit misleading.  Also, why is the RNA World label (the one in Jurassic Park lettering) near the beta = 0 curve if you expect large beta to be common (line 183). Perhaps it is better to move the label to the upper left of the axes.

Line 199. Can you put a into RNA context, please?

Line 210: should be commented out of the LaTeX.

Line 243: qualitatively similar? Explain.

Line 259: You actually calculate an estimate of R, under the assumption that there is no variation in the population. You don’t calculate R, per se.

Line 304: Here you’ve said that 0<=x<=1, but it’s not clear to me that x* in eqns 7, 9, 11 stay within those bounds. In fact, it seems from Figs 2 and 3 that x* sometimes falls outside the unit interval. Presumably if x* < 0 then we would actually say that x* = 0, but that should be clarified in the eqns.

Note on citations: I suspect the authors have typed, “( \cite{nickname1}, \cite{nickname2} )” leading to citations that look like, “( [33], [42] ).”  Using only the cite control sequence without round parenthesis will fix that. Also, using the sort&compress option with \usepackage in the preamble will shorten up citations like line 33.

I noticed a few typos in the References at the end of the paper.

Author Response

This paper provides a basic overview of how kin-selection thinking might benefit discussions surrounding the RNA World and the origins of life. The authors suggest that, by framing our thinking in terms of kin selection, we are better able to bring discoveries from other corners of evolutionary biology to bear on the problems presented by the RNA World. The take-home message is that kin-selection theory is one possible means by which advances can be achieved.

The authors’ modelling starts from the reasonable premise that early replicators were faced with a dilemma. They could either boost the replication rate of others through enzymatic activity, or allow their own replication to be boosted by acting as a template. Increased amounts of time spent boosting others is viewed as more “altruistic,” while increased amounts of time being boosted is viewed as more “selfish.” I understand that “altruism” would need to be present in sufficient quantities in order fuel the hypercycles one might be interested in studying.

The modelling continues by employing kin-selection methods to study simple models for the evolution altruism---i.e. the fuel for the origin of life. The authors’ kin-selection analysis produces three clear results in very short order (that results are easy to derive is a strength of the approach). Specifically,

(1)    The extent to which a replicator can share in the effects of its own altruism will be an important determinant of the evolution of altruism.

(2)    The extent to which replicators within a lineage compete with one another for resources will be another important determinant, with increased self-shading hindering altruism.

(3)    The details of how replicators move around in the environment will further complicate 1 and 2.

Results 1-3 will be familiar to those already well-versed in kin-selection theory, but the fact that they are formulated in terms of the RNA World bridges an important gap, in my view. Given the Aims of the journal, I believe readers will find the contribution interesting.

***Thank you for your kind words, and careful reading of the manuscript. We believe your suggestions greatly improved the manuscript.

Still, like any reviewer in the life sciences, I think the submission can be improved. To that end, I’ve offered suggestions for improvement, below.

Main Suggestions:

A.      The models used in the paper are likely to be viewed by the readership as being non-standard. The authors acknowledge this in at least one place (line 241). To really sell the idea that kin-selection theory can make a valuable contribution, I think it is important to apply it to models that will be more familiar to the Life community. I will concede that my suggestion may not be practicable at this stage, and that the absence of an application to lattice-based populations is not a fatal weakness. Bringing the methodology to bear on a model that this community really cares about is certainly something the authors should tackle soon, though.

*** We agree that it would be interesting and valuable to explicitly compare predictions of kin selection models and lattice based simulations, for example by actually measuring kin selection parameters such as relatedness in a simulation. We believe this is beyond the scope of this paper, which we see as an introduction to the approach, but we hope that the addition of Table 1 (l. 394) sets the stage for such future work. We thank reviewer 1 for this suggestion.

B.      I agree with the authors that there is something to be gained by viewing the RNA world through the kin selection lens. However, I don’t think they have made a solid case for this. Instead, they simply make vague assertions like that aroung line 333. Specifics, there, are important given the goal of the paper. Can the authors provide an example of how kin-selection thinking might point to new horizons in RNA World work? Another missed opportunity (but not the only one) occurs at line 178. There, the authors mention a connection to some work by Steve Frank, but they don’t elaborate. Bridging the gap is going to require more than simply providing a citation!

***We agree with the reviewer, and found this suggestion very helpful. In addition to making more explicit links to Frank’s work at l. 193, we have made explicit links to other models in l.193-196, l.212-214, l. 227-232, and l.286-291, as well as adding Table 1 (l.394) to aid the reader. Finally, we have added a subsection`Testability’ to the discussion, which fleshes out the kinds of new horizons we believe the kin selection approach points to. 

C.      A bit more detail on what the kin-selection methodology is about might be useful given the context. After poring over the paper, a reader could be forgiven if he/she claimed that kin-selection is just about swapping out derivatives for relatedness coefficients. But it’s way more than that! For example, the authors may wish to describe kin-selection methodology as a tool for reducing the state space of the evolving system.  By this I mean that the kin-selection methodology uses relatedness coefficients to effectively group different micro-states that, themselves, capture different configurations of genotypes. In other words, one needn’t track all the details about which genes go where; instead one can just use R to average. The basic idea of reducing the number of dimensions of a dynamical system would likely be familiar to a crowd used to seeing lattice-based models.

*** Very good point. We have expanded the section on relatedness (l. 98-114), which we believe addresses this issue.

Minor Suggestions:

Line 40: Hamilton reference is the wrong style.

***Thank you, changed

Line 66: “avoiding”

***Thanks, changed

Line 76: I did not get a good sense of what x was at this point, nor did I realize that it was bounded between 0 and 1 until Line 304.

***At this point X is not bounded, which we have now clarified (l.79-81), and we point out when it becomes bounded (l.153).

Line 96: “Relatedness captures the likelihood that two individuals share genes.” I am having difficulty understanding this for RNAs. Are related RNAs simply identical sequences? Please clarify. Also this might be a convenient place to address C, above.

***Good point, as mentioned above we now elaborate on this in l.98-114.

Line 104. I don’t agree that R is formally equivalent to dy/dx. Using R in place of dy/dx is a little heuristic introduced by Taylor and Frank (1996). Formally, R is a covariance divided by a variance (Michod & Hamilton, 1980).

*** Agreed. What we meant was R was formally the derivative in this model, not generally. We have clarified and elaborated (l.98-114). 

Line 106: “ … for cooperation to evolve …” Is the idea that, without cooperation, there is no life? Can you clarify why x*>0 is particularly important?

*** We believe there are two separate questions here. One is whether cooperation was necessary at the origin of life. We discuss why this is likely in the intro, but it’s not within the scope of this paper (or, so far, any paper) to prove that cooperation was necessary for the origin of life. We take the stance that cooperation was necessary, at various stages (as discussed in the intro, and covered more in depth in Higgs and Lehman 2015, Maynard Smith and Szathmary 1995), and we explore the conditions under which cooperation would evolve. The second question is why x*>0 is important, which is because by definition x*>0 is cooperation. We have clarified  this (l.115-116).

Line 150. I’m having a difficult time with the term “cooperative enzyme,” because I keep thinking we’re talking about cooperative binding like with Hb (are we?). If “cooperative enzyme” really refers to a replicator with greater x, then is it better to use “altruistic.” I know that “altruism” has gone out of style, but it might be useful to resurrect it here to avoid other kinds of confusion.

*** Yes, the reviewer makes a good point that we are looking at altruism in the strict sense. However, we have maintained the use of the term cooperation because it is standard in this field (e.g. Higgs and Lehman 2015), and altruism is a form of cooperation. We have clarified this link in l. 147-149. 

Line 150: delete the ref to equation 4, and make it clear that beta is a number between 0 and 1.

***Changed, l.161

Around eqn 7: Make it clear that this is a *candidate* ESS (you’ve only satisfied the first-order, necessary condition).

***Thank you, added (l.178).

Figure 2: this plot illustrates eqn 7 but only in the case where R is independent of x and beta. As you say later in the paper, R might depend on these things (and possibly more). I think that discussion needs to come earlier in the paper; otherwise Figure 2 might be a bit misleading.  Also, why is the RNA World label (the one in Jurassic Park lettering) near the beta = 0 curve if you expect large beta to be common (line 183). Perhaps it is better to move the label to the upper left of the axes.

***Thank you for pointing this out, we didn’t mean for the label to represent a specific region of parameter space, and we have deleted it to avoid confusion. Figure 2 is a representation of the results of the model in equation 4, which does implicitly assume that R is independent of the model parameters, which is standard practice in social evolution theory. We feel introducing this discussion earlier in the paper, before we allow R to depend on model parameters, would be confusing. 

Line 199. Can you put a into RNA context, please?

*** We believe we now do so in l. 208-223, l. 227-232, and l.235-238. 

Line 210: should be commented out of the LaTeX.

***Thanks! Changed

Line 243: qualitatively similar? Explain.

*** Yes, we mean qualitatively, and have added in l. 270. 

Line 259: You actually calculate an estimate of R, under the assumption that there is no variation in the population. You don’t calculate R, per se.

*** Good point, this wasn’t clear, and we have now changed l. 293.

Line 304: Here you’ve said that 0<=x<=1, but it’s not clear to me that x* in eqns 7, 9, 11 stay within those bounds. In fact, it seems from Figs 2 and 3 that x* sometimes falls outside the unit interval. Presumably if x* < 0 then we would actually say that x* = 0, but that should be clarified in the eqns.

*** Good point, we should be explicit about this, and have added l. 180. 

Note on citations: I suspect the authors have typed, “( \cite{nickname1}, \cite{nickname2} )” leading to citations that look like, “( [33], [42] ).”  Using only the cite control sequence without round parenthesis will fix that. Also, using the sort&compress option with \usepackage in the preamble will shorten up citations like line 33.

***Thank you, we have gotten rid of the brackets, but not used the sort&compress package as the full list follows the journal’s formatting guidelines. 

I noticed a few typos in the References at the end of the paper.

***Thank you, we believe all have been fixed. 

Round 2

Reviewer 2 Report

In this submission, Levin and West present revisions to their manuscript, in which they incorporate references mentioned in the original reviews. The manuscript is expanded with a qualitative description of how their model might relate to RNA world scenarios. A qualitative comparison of their model to existing models is made.

The revised manuscript has sufficiently addressed my concerns, and I recommend publication.